# QIVPT: QUANTUM-INSPIRED VISUAL PROMPT TUNING VIA GIVENS ROTATIONS-BASED ORTHOGONAL TRANSFORMATION

## ABSTRACT

Prompt tuning has become a powerful and parameter-efficient approach for adapting pre-trained vision transformers to diverse downstream tasks. However, existing methods often lack structural interpretability, treating prompt tokens as free parameters without semantic constraints. Inspired by the principles of quantum computation, particularly superposition and unitary evolution, we propose a novel method called Quantum-Inspired Visual Prompt Tuning (QIVPT). By viewing prompt tokens as semantic analogues of quantum states in a high-dimensional Hilbert space, we design a structured and orthogonality-preserving transformation over prompts using learnable sequences of Givens rotations. This approach simulates quantum-like semantic evolution, enabling controllable and reversible interactions among prompt embeddings, while maintaining low parameter overhead and enhancing adaptability across diverse tasks. Unlike parameterized quantum circuits, our method is lightweight, fully differentiable, and hardware-friendly. Extensive experiments on VTAB-1k and FGVC benchmarks demonstrate that QIVPT consistently outperforms existing prompt tuning baselines, offering improved generalization with minimal additional parameters.

## 1 INTRODUCTION

Prompt tuning has recently emerged as a compelling and parameter-efficient paradigm for adapting large pre-trained models to diverse downstream tasks. Originally developed in the context of natural language processing (NLP), prompt tuning has been successfully extended to vision-language and pure visual models (Lester et al., 2021; Li & Liang, 2021), particularly in the form of visual prompt tuning (VPT) (Jia et al., 2022a). This approach adapts pre-trained vision transformers (ViTs) by prepending a small set of learnable prompt tokens to the input sequence, while keeping the backbone model fixed. Despite requiring only a tiny fraction of the total parameters to be optimized, VPT has demonstrated surprisingly strong performance across a variety of tasks.

However, the underlying mechanism by which prompt tokens capture task-specific semantics and modulate internal representations remains poorly understood (Liu et al., 2022). Current approaches generally treat prompt tokens as unconstrained free parameters that are optimized solely for downstream performance, without incorporating inductive biases or structural constraints. Such designs may limit interpretability and lead to inefficiencies in token usage or interaction modeling. This raises a central challenge: how can we design prompt transformations that are simultaneously effective, structured, and semantically meaningful (Zhou et al., 2022b;a)?

Recently, Visual Fourier Prompt Tuning (VFPT) (Zeng et al., 2024) has taken a step in this direction by constraining prompt updates to the Fourier basis. VFPT shows that imposing a frequency-domain structure on prompts can improve generalization and efficiency, highlighting the benefit of embedding explicit mathematical priors into prompt design. However, Fourier transforms represent only a specific family of orthogonal transformations, leaving open the question of whether a more general and flexible framework can be developed.

Hereby, quantum computing provides a useful source of inspiration. In quantum mechanics, states are represented in a high-dimensional Hilbert space and evolve under unitary operators that preserve structure and encode global interactions (Preskill, 2018). A remarkable property is that any unitary

operator, no matter how complex, can be decomposed into a sequence of Givens rotations (Arrazola et al., 2022)—each acting only on a two-dimensional subspace but together forming intricate global dynamics (Lloyd, 1996; Shende et al., 2006; Nielsen & Chuang, 2010). This principle suggests that rich transformations can emerge from the repeated composition of simple, structured local rotations.

Motivated by this view, we revisit prompt tuning from a structural perspective. Instead of allowing prompt tokens to update in an unconstrained way, we let them evolve through a sequence of learnable Givens rotations. Each rotation governs how two prompt dimensions interact, and the composition of many such rotations enables complex global behaviors. In this way, the learning process of prompt tokens resembles a form of controlled semantic evolution: global expressivity is built up gradually from simple, interpretable local transformations (Liu et al., 2022; Zhou et al., 2022a; Nie et al., 2023).

While we initially explored parameterized quantum circuits as a more direct implementation, they face severe practical issues—including computational cost, simulator inefficiency, and gradient instability—that hinder large-scale deployment. To retain the appealing principles of orthogonality, reversibility, and global interaction without incurring the heavy overhead of parameterized quantum circuits, we adopt structured orthogonal transformations as a lightweight and scalable alternative (Lloyd, 1996; Shende et al., 2006; Nielsen & Chuang, 2010).

Specifically, we propose a learnable Givens Rotation Layer that applies a sequence of such rotations to prompt embeddings. These transformations are differentiable, computationally efficient, and inherently interpretable, making them well suited for modeling semantic interactions among prompt tokens. Importantly, we show that Fourier transforms can be viewed as a special case of Givens rotations compositions. Thus, our method— Quantum-Inspired Visual Prompt Tuning (QIVPT)—naturally generalizes VFPT by expanding from the Fourier subspace to the full orthogonal group (Zeng et al., 2024; Dong et al., 2024b).

The main contributions of this study are summarized as follows:

- We introduce Quantum-Inspired Visual Prompt Tuning (QIVPT), a novel framework that constrains prompt evolution through learnable Givens rotations. This design incorporates explicit orthogonal structure, offering interpretability and efficiency while preserving the flexibility of prompt tuning. Notably, Visual Fourier Prompt Tuning (VFPT) arises as a special case within our framework, highlighting that QIVPT naturally generalizes it from the Fourier basis to the full orthogonal group.

- We realize QIVPT by proposing a lightweight and scalable Givens Rotation Layer, which composes local two-dimensional rotations into expressive global transformations, retaining the desirable properties of orthogonality, reversibility, and norm preservation.

- We conduct extensive experiments on VTAB-1k and FGVC benchmarks, demonstrating that QIVPT consistently outperforms existing prompt tuning methods in both accuracy and parameter efficiency across diverse tasks.

## 2 RELATED WORK

**Prompt Tuning in Vision.** Prompt tuning has recently gained traction in vision-language models (VLMs) (Zhou et al., 2022b;a; Jia et al., 2022b; Zhang et al., 2022a; Han et al., 2023; Nie et al., 2023) as a lightweight alternative to full fine-tuning. The central idea is to inject a small set of learnable tokens (prompts) into the input sequence of vision transformers (ViTs), enabling task-specific adaptation while keeping the backbone frozen. Early works treat these prompt tokens as free-form vectors optimized via downstream supervision (Jia et al., 2022b). Subsequent variants propose visual prompt tuning (VPT) (Jia et al., 2022b), prompt ensembling (Zhou et al., 2022b;a), Fourier-based prompt tuning (Zeng et al., 2024), and prompt tuning for fine-grained tasks (Liu et al., 2023). However, these methods often lack structural constraints or interpretability. Our work differs by introducing a quantum-inspired structural prior via orthogonal transformations, enhancing both semantic controllability and interpretability in the prompt space.

**Orthogonal and Structured Transformations.** Orthogonal transformations are widely used in deep learning to preserve norm and stability (Saxe et al., 2014; Bansal et al., 2018), and have been

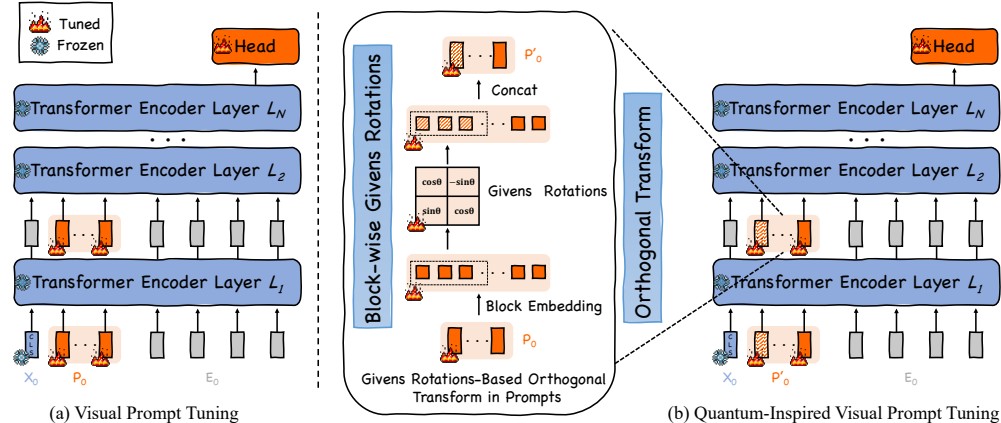

Figure 1: Overview of VPT vs. QIVPT (ours) frameworks. (a) Original Visual Prompt Tuning. (b) The overall architecture of our proposed QIVPT (see Sec. 3.2).

explored in low-rank adaptation (He et al., 2022), normalization (Daneshmand et al., 2021), and recurrent networks (Arjovsky et al., 2016; Wisdom et al., 2016). Recently, Givens rotations have gained attention as efficient, differentiable building blocks for learning orthogonal matrices (Fawzi et al., 2019; Choromanski et al., 2021; Li et al., 2020). Compared to full orthogonal parameterizations like Householder reflections or Cayley transforms (Lezcano-Casado & Martínez-Rubio, 2019), Givens rotations provide fine-grained, low-rank control with fewer parameters and lower computational cost. In our work, we adopt Givens rotations to build a lightweight, learnable transformation layer over prompt tokens. This design preserves orthogonality while allowing semantic entanglement and disentangled control in the prompt space.

**Quantum-Inspired Learning.** Quantum computing principles such as superposition and entanglement have inspired various machine learning frameworks (Schuld et al., 2015; Beer et al., 2020; Kong et al., 2022; Gao et al., 2022), including quantum kernel learning (Havlíček et al., 2019), variational quantum circuits (Mari et al., 2020; Mitarai et al., 2018), and quantum-inspired classical models (Zhang et al., 2022b; Casas & Cervera-Lierta, 2023). These methods view data representations as high-dimensional quantum states undergoing unitary evolution (Lloyd, 1996; Nielsen & Chuang, 2010). Motivated by this analogy, recent works explore simulating quantum behavior via classical structures (Shen et al., 2023; Zhao et al., 2023; Barthe & Pérez-Salinas, 2024; Wang et al., 2025). However, directly applying parameterized quantum circuits to deep models often suffers from scalability and training instability. Our method avoids these issues by emulating quantum dynamics using structured orthogonal transformations (Givens rotations) that are efficient, interpretable, and hardware-agnostic. To our knowledge, we are the first to apply quantum-inspired orthogonal evolution to prompt tuning in visual transformers.

## 3 METHODOLOGY

In this section, we present Quantum-Inspired Visual Prompt Tuning (QIVPT), a prompt tuning scheme that applies learnable, Givens rotations–based orthogonal transformations to the prompt embedding space. By emulating the norm-preserving dynamics of unitary evolution, QIVPT induces structured and reversible interactions among prompt dimensions, thereby improving expressivity and task adaptivity without modifying the frozen backbone. An overview and comparison with standard Visual Prompt Tuning (VPT) are shown in Fig. 1.

We begin by reviewing the formulation and limitations of VPT in Sec. 3.1. We then introduce QIVPT in Sec. 3.2, where layerwise compositions of Givens rotations yield a lightweight, interpretable, and scalable prompt transformation. Finally, in Sec. 3.3, we show that Fourier-based prompt tuning (VFPT) emerges as a special case within the broader family of orthogonal transforms realized by QIVPT.

## 3.1 Background of Original Visual Prompt Tuning

Visual Prompt Tuning (VPT) (Jia et al., 2022a) is a parameter-efficient paradigm for adapting large-scale vision transformers (ViTs) to downstream tasks without updating backbone weights. Inspired by prompt tuning in NLP, VPT introduces a small number of learnable prompt tokens into the input sequence. These tokens act as task-specific instructions, guiding the pretrained transformer to adapt to new domains while keeping the backbone frozen.

As shown in Fig. 1 (a), given an image $\mathcal{X}$, ViTs tokenize it into patch embeddings $\mathbf{E}_0 \in \mathbb{R}^{m \times d}$, where $m$ is the number of image patches and $d$ is the embedding dimension. Let $\mathbf{x}_0 \in \mathbb{R}^{1 \times d}$ denote the [CLS] token and $\mathbf{P} \in \mathbb{R}^{p \times d}$ be a set of $p$ learnable prompt tokens. The modified input to the first transformer layer is

$$[\mathbf{x}_1, \mathbf{Z}_1, \mathbf{E}_1] = L_1([\mathbf{x}_0, \mathbf{P}, \mathbf{E}_0]), \tag{1}$$

where $\mathbf{Z}_1 \in \mathbb{R}^{p \times d}$ denotes the prompt features after the first layer. The concatenated sequence $[\mathbf{x}_1, \mathbf{Z}_1, \mathbf{E}_1] \in \mathbb{R}^{(1+p+m) \times d}$ is then passed through the remaining transformer blocks.

● and ● indicate learnable and frozen parameters, respectively. Notably for ViTs, since positional embeddings are added before prompt insertion, the relative position of prompt tokens is invariant; hence $[\mathbf{x}_0, \mathbf{P}, \mathbf{E}_0]$ and $[\mathbf{x}_0, \mathbf{E}_0, \mathbf{P}]$ are equivalent.

**VPT-Shallow**  Prompt tokens are inserted only at the input of the first transformer block $L_1$ and remain unmodified thereafter. For $i = 2, \dots, L$,

$$[\mathbf{x}_i, \mathbf{Z}_i, \mathbf{E}_i] = L_i([\mathbf{x}_{i-1}, \mathbf{Z}_{i-1}, \mathbf{E}_{i-1}]), \tag{2}$$

$$\mathbf{y} = \texttt{Head}(\mathbf{x}_L). \tag{3}$$

**VPT-Deep**  Prompt tokens are introduced at every transformer layer. For the $i$-th layer, a separate set of learnable prompts $\mathbf{P}_{i-1} \in \mathbb{R}^{p \times d}$ is inserted:

$$[\mathbf{x}_i, \mathbf{Z}_i, \mathbf{E}_i] = L_i([\mathbf{x}_{i-1}, \mathbf{P}_{i-1}, \mathbf{E}_{i-1}]). \tag{4}$$

Although VPT is simple and effective, it still exhibits several limitations. Its interaction mechanism lacks structural guidance, as the prompt tokens are learned as free vectors without any inductive bias or internal organization. Moreover, the reliance on a fixed embedding basis introduces a coordinate bias, which may restrict expressiveness and hinder generalization to unseen domains. Finally, the influence of prompts on attention dynamics and downstream predictions remains opaque, making the adaptation process difficult to interpret or control. These observations motivate a more structured, interpretable, and theoretically grounded formulation, cf. Sec. 3.2.

## 3.2 Quantum-Inspired Visual Prompt Tuning

We propose Quantum-Inspired Visual Prompt Tuning (QIVPT), a lightweight and theoretically motivated variant that leverages orthogonal transformations—specifically Givens rotations—to model structured prompt evolution, cf. Fig. 1 (b).

**Givens Rotations**  Let the prompt token matrix be $\mathbf{P} \in \mathbb{R}^{p \times d}$. A Givens rotation $G(i, j, \theta) \in \mathbb{R}^{d \times d}$ rotates vectors within the $(i, j)$-plane by angle $\theta \in [-\pi, \pi]$:

$$\mathbf{G}(i, j, \theta) = \mathbf{I} + (\cos\theta - 1)(\mathbf{e}_i \mathbf{e}_i^\top + \mathbf{e}_j \mathbf{e}_j^\top) + \sin\theta(\mathbf{e}_j \mathbf{e}_i^\top - \mathbf{e}_i \mathbf{e}_j^\top), \tag{5}$$

where $\mathbf{e}_i, \mathbf{e}_j \in \mathbb{R}^d$ are standard basis vectors. Composing $K$ such rotations yields an orthogonal transform

$$\mathbf{Q} = \prod_{k=1}^{K} \mathbf{G}(i_k, j_k, \theta_k), \qquad \mathbf{Q}^\top \mathbf{Q} = \mathbf{I}. \tag{6}$$

Additional background on orthogonal transformations in quantum circuits is provided in Appendix A.

**Orthogonal Prompt Transformation**  We transform the selected prompt rows along the embedding dimension via right-multiplication by $\mathbf{Q}$. Let $r \in \{0, \ldots, p\}$ be the number of transformed prompts and $\alpha \triangleq r/p$. Denote the first $r$ prompt rows by $\mathbf{P}_{1:r} \in \mathbb{R}^{r \times d}$ and the remaining by $\mathbf{P}_{r+1:p} \in \mathbb{R}^{(p-r) \times d}$. We apply

$$\textcolor{red}{\mathbf{P}_{\mathcal{Q}}} \;=\; \mathbf{P}_{1:r}\,\mathbf{Q}, \qquad \hat{\mathbf{P}} \;=\; \begin{bmatrix} \textcolor{red}{\mathbf{P}_{\mathcal{Q}}} \\ \mathbf{P}_{r+1:p} \end{bmatrix} \in \mathbb{R}^{p \times d}. \tag{7}$$

To ensure stability during training, each angle is parameterized by

$$\theta_k \;=\; \pi \tanh(\phi_k), \quad \phi_k \in \mathbb{R}, \tag{8}$$

which bounds $\theta_k \in (-\pi, \pi)$.

**Integration with Transformer Layers**  We apply independent orthogonal transformations at each transformer layer $L_i$ to obtain layer-specific prompts $\hat{\mathbf{P}}_{i-1}$:

$$[\mathbf{x}_i, \mathbf{Z}_i, \mathbf{E}_i] = \textcolor{blue}{L_i}\big([\mathbf{x}_{i-1}, \hat{\mathbf{P}}_{i-1}, \mathbf{E}_{i-1}]\big), \quad i = 1, \ldots, L, \tag{9}$$

$$\mathbf{y} = \texttt{Head}(\mathbf{x}_L), \tag{10}$$

where $\mathbf{E}_i \in \mathbb{R}^{m \times d}$ and $\mathbf{x}_i \in \mathbb{R}^{1 \times d}$.

**Block-wise Efficiency in High Dimensions**  For large $d$, we adopt a block-wise strategy: partition the hidden dimension into blocks of size 16 and apply Givens rotations independently within each block, preserving orthogonality and expressivity while reducing computation and parameters. We further study the trade-off between performance and efficiency under different block sizes, and report detailed results in Appendix F.

**Interpretation and Benefits**  QIVPT induces *semantic entanglement* among prompt dimensions via norm-preserving rotations, yielding several advantages:

- **Interpretability.** Because each transformation is orthogonal, both vector norms and inner products are preserved. This means the geometry of the embedding space remains intact, allowing prompt evolution to be understood as a sequence of controlled rotations rather than arbitrary updates. As a result, we obtain a clear geometric interpretation of how prompts interact and evolve across layers, making the adaptation process more transparent compared to unconstrained prompt tuning.
- **Lightweight design.** Each Givens rotation introduces only a single learnable angle parameter. To transform a high-dimensional embedding, only a small number of such rotations are needed, making the additional parameter count negligible relative to the full model. Moreover, the rotations are implemented as simple two-dimensional updates, which are computationally inexpensive and easy to parallelize.
- **Scalability.** In practice, directly applying rotations in very high dimensions may be costly. To address this, QIVPT adopts a block-wise strategy that partitions the embedding dimension into smaller blocks (e.g., size 16) and applies rotations independently within each block. This design substantially reduces complexity while still capturing rich local interactions, ensuring the method scales efficiently to large models and long prompt sequences.

Compared to unconstrained prompt updates, QIVPT enforces structured orthogonal dynamics that enhance both interpretability and empirical performance (see Sec. 4).

## 3.3 Givens Rotations as a Generalization of Fourier Transform

Visual Fourier Prompt Tuning (VFPT) constrains prompt updates within the Fourier basis, injecting a frequency-domain inductive bias. While effective, the Fourier transform is only one member in the broader family of orthogonal/unitary transforms. In this subsection we formalize that viewpoint: *complex* Givens rotations (plus diagonal phase factors) generate $\mathbf{U}(n)$, and the DFT is therefore a special case. Under the real embedding, this implies that VFPT sits inside the orthogonal-transform family realized by QIVPT. We defer detailed proofs and additional factorization examples to Appendix B.

**Complex Givens rotations**  For $n \geq 2$, a complex Givens rotation $\mathbf{G}(i, j, \theta, \phi) \in \mathbb{C}^{n \times n}$ acts as identity except on the $(i, j)$-plane, where

$$\mathbf{G}(i, j, \theta, \phi)\big|_{\mathrm{span}\{\mathbf{e}_i, \mathbf{e}_j\}} = \begin{bmatrix} \cos\theta & -\sin\theta\, e^{-i\phi} \\ \sin\theta\, e^{i\phi} & \cos\theta \end{bmatrix}, \qquad \theta \in [0, \tfrac{\pi}{2}], \ \phi \in [0, 2\pi).$$

This block is unitary and $\ell_2$-norm preserving.

**Lemma 1** (Givens–QR generates $\mathbf{U}(n)$). *Let $\mathbf{U} \in \mathbf{U}(n)$. There exist complex Givens rotations $\mathbf{G}(i_t, j_t, \theta_t, \phi_t)$ and diagonal unitary matrices $\mathbf{D}_L, \mathbf{D}_R$ (phase factors) such that*

$$\mathbf{U} = \mathbf{D}_L \Big( \prod_{t=1}^{T} \mathbf{G}(i_t, j_t, \theta_t, \phi_t) \Big) \mathbf{D}_R.$$

*Proof.* Perform Givens-based QR elimination column-wise: left-multiply by $\mathbf{G}(k-1, k, \theta, \phi)$ to zero subdiagonals while preserving unitarity, yielding $\mathbf{G}_T \cdots \mathbf{G}_1 \mathbf{U} = \mathbf{R}$ upper-triangular and unitary. An upper-triangular unitary is diagonal with unimodular entries, hence $\mathbf{R} = \mathbf{D}$ (diagonal phase). Thus $\mathbf{U} = (\mathbf{G}_T \cdots \mathbf{G}_1)^{-1}\mathbf{D}$, and phases can be absorbed to both sides.

**Theorem 1** (DFT factors into complex Givens rotations). *Let $\mathbf{F}_n \in \mathbb{C}^{n \times n}$ denote the $n$-point DFT with $[\mathbf{F}_n]_{k\ell} = \frac{1}{\sqrt{n}} \omega_n^{(k-1)(\ell-1)}$. Then there exist angles $\{\theta_t, \phi_t\}$, indices $\{i_t, j_t\}$, and diagonal phase matrices $\mathbf{D}_L, \mathbf{D}_R$ such that*

$$\mathbf{F}_n = \mathbf{D}_L \Big( \prod_{t=1}^{T} \mathbf{G}(i_t, j_t, \theta_t, \phi_t) \Big) \mathbf{D}_R.$$

*Moreover, under the real embedding*

$$\widehat{\mathbf{U}} := \begin{bmatrix} \Re\mathbf{U} & -\Im\mathbf{U} \\ \Im\mathbf{U} & \Re\mathbf{U} \end{bmatrix},$$

$\widehat{\mathbf{F}_n}$ *is a real* orthogonal *matrix that factors into a product of (real) Givens rotations.*

*Proof.* By Lemma 1, since $\mathbf{F}_n$ is unitary, it admits the stated factorization. The mapping $\widehat{\cdot}$ sends unitary to real orthogonal matrices because $\widehat{\mathbf{U}}^\top \widehat{\mathbf{U}} = \mathbf{I}$ whenever $\mathbf{U}^*\mathbf{U} = \mathbf{I}$. Each complex $2 \times 2$ rotation block becomes a $4 \times 4$ real orthogonal block, which (up to permutation) is a product of two standard real Givens rotations.

*Remark* (Connection to FFT butterflies). After extracting a diagonal phase, each complex Givens block coincides with a 2-point butterfly $\frac{1}{\sqrt{2}} \begin{bmatrix} 1 & \omega \\ 1 & -\omega \end{bmatrix}$ with twiddle $\omega$. Hence the layered FFT factorization is a structured choice of the Givens–phase factors in Theorem 1.

## 4  EXPERIMENTS

### 4.1  SETUP

**Datasets**  We evaluate QIVPT on two standard image classification benchmarks: (1) **VTAB-1k**, a collection of 19 tasks grouped into Natural, Specialized, and Structured categories, each with 1000 training samples. We use standard train/val/test splits and report average test accuracy over three runs; (2) **FGVC**, including CUB-200-2011, NABirds, Oxford Flowers, Stanford Dogs, and Stanford Cars.

**Backbones & Baselines**  Our experiments are conducted on the ViT-Base/16 (Dosovitskiy et al., 2021) backbone pretrained on supervised ImageNet-21k. We compare QIVPT with a broad spectrum of adaptation strategies. These include: *full fine-tuning* and *linear probing* (Iofinova et al., 2022), *partial layer tuning* (Yosinski et al., 2014), and *MLP tuning* (Chen et al., 2020). Parameter-efficient approaches are also covered, such as Side-Tuning (Zhang et al., 2020), Bias/BitFit (Rebuffi et al., 2017), Adapter (Cai et al., 2020), LoRA (Hu et al., 2021), AdaptFormer (Chen et al., 2022), and ARC$_{\mathrm{att}}$ (Dong et al., 2024a). Finally, we include prompt-based methods: Visual Prompt Tuning (VPT) (Jia et al., 2022b), EXPRES (Das et al., 2023), E2VPT (Han et al., 2023), Visual Fourier Prompt Tuning (VFPT), and our proposed QIVPT.

Table 1: **Image classification accuracy for ViT-Base/16** pretrained on supervised ImageNet-21k. Following Jia et al. (2022b); Han et al. (2023), we report the average test accuracy (three runs) on VTAB-1k (Zhai et al., 2019) and FGVC (Jia et al., 2022b) benchmarks, and "Number of Wins" in [·] compared to full fine-tuning (Full). ▷ denotes the method with highest "Number of Wins" compared to Full. We further report "Number of Wins to VFPT" in {·}. "Tuned/Total" is the average percentage of tuned parameters required by 24 tasks. "Scope" indicates the tuning scope of each method. "Additional parameters" is the existence of parameters in addition to the pretrained backbone and linear head. **Bold** and **Underline** indicate the best and the second best results. QIVPT outperforms full fine-tuning in **22 of 24** instances with fewer trainable parameters and beats VFPT in **24 of 24** cases with same parameters. † denotes methods using soft filtered prompts to reduce the parameter usage in learnable visual prompts, requiring specialized devices to facilitate acceleration. Per-task results are available in Appendix D.1. Same for Table 2.

| ViT-Base/16 (85.8M) | Tuned/ Total | Scope Input | Backbone | Extra params | VTAB-1k [19] Natural [7] | Specialized [4] | Structured [8] | Mean Total | FGVC [5] |
|---|---|---|---|---|---|---|---|---|---|
| Full (2022) | 100.00% | | ✓ | | 75.88% | 83.36% | 47.64% | 65.57% | 88.54% |
| Linear (2022) | 0.08% | | | | 68.93% [1] | 77.16% [1] | 26.84% [0] | 52.94% | 79.32% [0] |
| Partial-1 (2014) | 8.34% | | | | 69.44% [2] | 78.53% [0] | 34.17% [0] | 56.52% | 82.63% [0] |
| MLP-3 (2020) | 1.44% | | | ✓ | 67.80% [2] | 72.83% [0] | 30.62% [0] | 53.21% | 79.80% [0] |
| Sidetune (2020) | 10.08% | | ✓ | ✓ | 58.21% [0] | 68.12% [0] | 23.41% [0] | 45.65% | 78.35% [0] |
| Bias (2017) | 0.80% | | ✓ | | 73.30% [3] | 78.25% [0] | 44.09% [2] | 62.05% | 88.41% [3] |
| Adapter (2020) | 1.02% | | ✓ | ✓ | 70.67% [4] | 77.80% [0] | 33.09% [0] | 62.41% | 85.46% [1] |
| LoRA (2021) | — | | ✓ | ✓ | 78.26% [5] | 83.78% [2] | 56.20% [7] | 72.25% | 89.46% [3] |
| AdaptFormer (2022) | — | | ✓ | ✓ | 80.56% [6] | 84.88% [4] | 58.83% [7] | 72.32% | |
| ARC$_{att}$ (2024a) | — | | ✓ | ✓ | 80.41% [7] | 85.55% [3] | 58.38% [8] | 72.32% | 89.12% [4] |
| VPT-S (2022b) | 0.16% | ✓ | | ✓ | 76.81% [4] | 79.66% [0] | 46.98% [4] | 64.85% | 84.62% [1] |
| VPT-D (2022b) | 0.73% | ✓ | | ✓ | 78.48% [6] | 82.43% [2] | 54.98% [8] | 69.43% | 89.11% [4] |
| EXPRES (2023) | — | ✓ | | ✓ | 79.69% [6] | 84.03% [3] | 54.99% [8] | 70.20% | — |
| † E2VPT (2023) | 0.39% | ✓ | ✓ | ✓ | 80.01% [6] | 84.43% [3] | 57.39% [8] | 71.42% | 89.22% [4] |
| VFPT (2024) | 0.66% | ✓ | | ✓ | 81.35% [6] | 84.93% [4] | 60.19% [8] | 73.20% | 89.24% [4] |
| ▷ QIVPT (Ours) | 0.66% | ✓ | | ✓ | **82.07%** [6] {7} | **86.20%** [4] {4} | **61.08%** [8] {8} | **74.10%** | **89.47%** [4] {5} |

**Training** Following VPT (Jia et al., 2022a) and VFPT (Jia et al., 2022b), we adopt a similar hyperparameter tuning strategy for QIVPT. Specifically, we perform a grid search over the learning rate (i.e., [5, 2.5, 1, 0.5, 0.25, 0.1, 0.05]) and weight decay (i.e., [0.01, 0.001, 0.0001]) on the validation set to select the best configuration. QIVPT is trained with a cosine decay learning rate schedule for 100 epochs under standard settings. This ensures a fair and consistent comparison while demonstrating that QIVPT achieves strong performance without relying on aggressive optimization tricks. The complete set of hyperparameters is provided in Appendix E.

## 4.2 MAIN RESULTS

In this section, we evaluate the effectiveness of QIVPT from three perspectives: task-level performance, architectural generalization, and qualitative visualization. Detailed per-task results are provided in Appendix D.

**Task-Level Results** Tab. 1 presents the results of fine-tuning a pre-trained ViT-B/16, averaged across four diverse downstream task groups, comparing QIVPT with 14 other tuning protocols. The results indicate that: QIVPT consistently achieves top-1 or top-2 accuracy across all task groups, including FGVC, Natural, Specialized, and Structured tasks. Specifically, QIVPT attains the highest mean accuracy of 74.10%, surpassing full fine-tuning by a significant margin of +8.53%, while requiring only 0.66% of total parameters to be updated. Moreover, QIVPT achieves 22 out of 24 wins over Full and 24 out of 24 wins over VFPT, establishing a new state-of-the-art in the trade-off between performance and efficiency. These results demonstrate that QIVPT not only matches or exceeds the task-level performance of existing methods, but does so with far fewer trainable parameters, confirming its strong generalization ability and practical applicability for real-world deployment.

**Architectural Generalization** We evaluate the architectural generalization ability of QIVPT on VTAB-1k (Zhai et al., 2019), using the Swin-Base (Liu et al., 2021) backbone pretrained on supervised ImageNet-21k. Swin utilizes local multi-head self-attention within shifted windows and hierarchically merges patch embeddings at deeper stages. Following prior works (Jia et al., 2022b; Han et al., 2023), we insert prompt tokens only within local windows and skip them during patch merging, ensuring a consistent and fair fine-tuning setup.

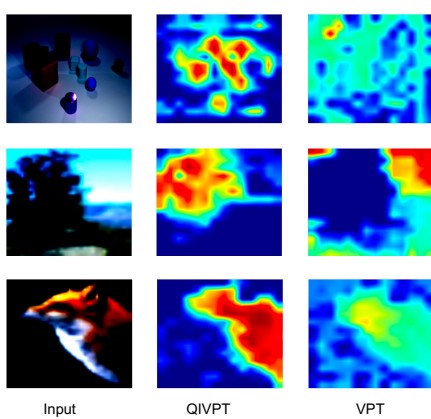

Input     QIVPT     VPT

Figure 2: GradCAM heatmap.

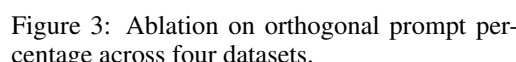

Figure 3: Ablation on orthogonal prompt percentage across four datasets.

As shown in Table 2, QIVPT achieves strong performance across all three subgroups of VTAB. Specifically, it attains 84.83% on Natural, 86.50% on Specialized, and 59.39% on Structured, outperforming VPT (Jia et al., 2022b) and VFPT (Zeng et al., 2024) with comparable or even fewer tunable parameters. Notably, QIVPT surpasses VFPT on all Natural, Specialized and Structured tasks, demonstrating improved robustness across both low-level and high-level domains. While full fine-tuning still provides the highest accuracy on Structured tasks, it requires

Table 2: **Image classification accuracy for Swin-Base** pretrained on supervised ImageNet-21k. Per-task results are available in Appendix D.2.

| Swin-Base (86.7M) | Tuned/ Total | VTAB-1k [19] | | |
| --- | --- | --- | --- | --- |
| | | *Natural* [7] | *Specialized* [4] | *Structured* [8] |
| Full (2022) | 100.00% | 79.10% | 86.21% | **59.65%** |
| Linear (2022) | 0.06% | 73.52% [5] | 80.77% [0] | 33.52% [0] |
| Partial-1 (2014) | 14.58% | 73.11% [4] | 81.70% [0] | 34.96% [0] |
| MLP-3 (2020) | 2.42% | 73.56% [5] | 75.21% [0] | 35.69% [0] |
| Bias (2017) | 0.29% | 74.19% [2] | 80.14% [0] | 42.42% [0] |
| VPT (2022b) | 0.25% | 76.78% [6] | 83.33% [0] | 51.85% [0] |
| † E2VPT (2023) | 0.21% | 83.31% [6] | 84.95% [2] | 57.35% [3] |
| VFPT (2024) | 0.27% | 84.53% [7] | 86.15% [2] | 58.21% [3] |
| ▷ QIVPT (Ours) | 0.27% | 84.83% [7] {7} | 86.50% [2] {4} | 59.39% [3] {8} |

tuning all model parameters. In contrast, QIVPT achieves competitive results with just 0.27% of parameters updated, reaffirming its effectiveness in parameter-efficient transfer under architectural variations.

**Visualization of QIVPT**   To better illustrate the differences in model attention, we provide Grad-CAM visualizations in Figure 2 to compare the attention patterns of VPT and our proposed QIVPT. In the first row, QIVPT captures a greater number of objects within the scene, indicating a broader and more comprehensive understanding of visual context. In the second row, QIVPT accurately focuses on the tree region, whereas VPT incorrectly attends to the sky, reflecting improved semantic alignment.In the third row, QIVPT places stronger emphasis on the bird's head (an essential discriminative region for fine-grained classification), demonstrating its superior ability to attend to task-relevant details.

## 4.3 ABLATION STUDIES

**Orthogonal Prompt Percentage**   We analyze the impact of varying the percentage of orthogonal prompts across datasets (Fig. 3). The optimal ratio is dataset-dependent: CIFAR-100 and DMLab peak at 70%, RESISC45 at 90%, while DTD performs best with only 30%. Datasets with higher visual diversity (e.g., RESISC45) favor larger proportions of orthogonal prompts, whereas simpler or texture-centric datasets (e.g., DTD) require fewer. This highlights the importance of adaptively tuning the ratio, confirming that orthogonal prompts naturally generalize VPT by providing complementary representational capacity.

**Orthogonal Prompt Dimension**   We study the dimensional axis along which the Givens-based orthogonal transformations are applied: the token sequence space, the hidden embedding dimension, or both simultaneously. As shown in Tab. 3 (a), applying transformations solely in the hidden dimension already provides substantial gains over the sequence axis, highlighting the importance of semantic modulation within the embedding space. Moreover, combining both sequence and hidden

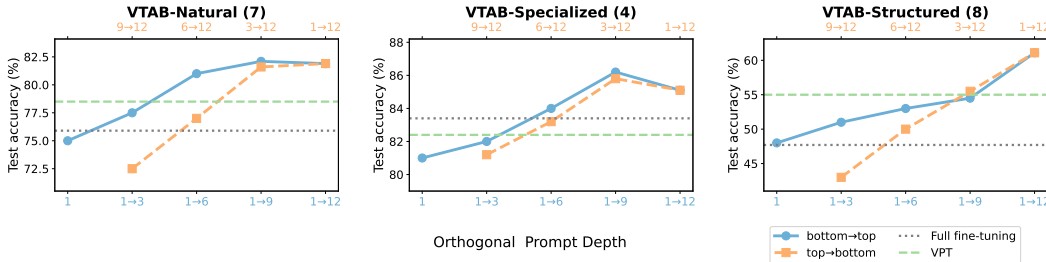

Figure 4: Ablation on orthogonal prompt depth. We select the best prompt length for each variant with validation sets. $i \rightarrow j$ indicates the Transformer layer indices where prompts are inserted, with the 1-st layer being the one closest to the input. ViT-B has 12 layers in total.

Table 3: Ablation on VTAB-1k. "Prompt Location" denotes where orthogonal prompts are inserted relative to original visual prompts. "Orthogonal Dim." indicates the dimension along which the transform is applied.

(a) Orthogonal Prompt Dimension

| Orthogonal Dim. | | VTAB-1k | |
| Sequence | Hidden | Natural | Specialized |
|---|---|---|---|
| ✓ | | 80.25 | 83.57 |
| | ✓ | 81.93 | 85.92 |
| ✓ | ✓ | **82.07** | **86.20** |

(b) Orthogonal Prompt Location

| Prompt | VTAB-1k | |
| Location | Natural | Specialized |
|---|---|---|
| $\mathcal{A}$ | 81.53 | 85.13 |
| $\mathcal{R}$ | 78.75 | 83.35 |
| $\mathcal{P}$ | **82.07** | **86.20** |

directions further improves performance, yielding the best results on both Natural and Specialized tasks. This indicates that the two axes capture complementary information, and jointly modeling them enhances the expressivity of orthogonal prompts.

**Orthogonal Prompt Location**   We investigate three strategies for inserting orthogonal prompts into the input sequence: $\mathcal{P}$ (Prepend), where prompts are inserted before patch tokens; $\mathcal{A}$ (Append), where prompts are placed after patch tokens; and $\mathcal{R}$ (Random), where prompts are interleaved at arbitrary positions. As reported in Tab. 3 (b), the $\mathcal{P}$ strategy outperforms the others on both Natural and Specialized splits of VTAB-1k, demonstrating that placing orthogonal prompts at the beginning of the sequence allows earlier transformer layers to better attend to and condition on them. In contrast, $\mathcal{A}$ and $\mathcal{R}$ reduce prompt visibility and weaken their semantic guidance.

**Orthogonal Prompt Depth**   Fig. 4 investigates the effect of orthogonal prompt insertion depth on QIVPT. Each variant reports the best prompt length chosen by validation sets. We observe that performance consistently improves as prompts are inserted deeper, with bottom→top injection yielding the best results around mid-to-deep layers (e.g., $1 \rightarrow 9$). In contrast, inserting prompts from top→bottom is less effective, especially when limited to shallow depths. This suggests that early-layer prompts play a more critical role in guiding adaptation, while orthogonal prompts at later layers offer diminishing benefits. Compared to both full fine-tuning and VPT baselines, QIVPT achieves higher accuracy across all VTAB categories with fewer trainable parameters.

## 5   CONCLUSIONS

In this work, we proposed Quantum-Inspired Visual Prompt Tuning (QIVPT), a novel framework that integrated structured orthogonal transformations into visual prompt tuning through learnable Givens rotations. By drawing inspiration from quantum mechanics, we interpreted prompt tokens as semantic states and evolved them via reversible, norm-preserving operations to enhance task adaptability and interpretability. Our Givens Rotation Layer offered a lightweight and parameter-efficient means of inducing semantic entanglement across prompt dimensions, without requiring access to quantum hardware. Extensive experiments on VTAB-1k and FGVC benchmarks demonstrated that QIVPT consistently outperformed conventional prompt tuning baselines. We believe this work provided a new perspective on structured prompt design and opened promising directions for quantum-inspired representation learning in vision.

ETHICS STATEMENT

This work complies with the ICLR Code of Ethics. All experiments are conducted on publicly available benchmark datasets (e.g., VTAB-1k and FGVC) under their respective licenses and usage guidelines. No private, proprietary, or personally identifiable information is involved, and no human subjects were recruited.

Our proposed method, QIVPT, aims to improve efficiency and interpretability in visual prompt tuning. The research does not introduce foreseeable misuse risks beyond those inherent to existing machine learning techniques. Nonetheless, as with all approaches relying on pretrained models, potential biases in the underlying datasets may propagate into downstream tasks. We encourage future work to further examine fairness, robustness, and broader societal impacts.

All authors affirm adherence to ethical research practices, including transparency, reproducibility, and respect for intellectual property.

REPRODUCIBILITY STATEMENT

All experiments in this paper were conducted on a single NVIDIA RTX 4090 GPU. The core implementation of QIVPT can be found in Appendix C, and the detailed hyperparameter configurations are provided in Appendix E. We have uploaded the full source code along with training scripts and instructions in the supplementary materials to ensure full reproducibility.

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

## APPENDIX SUMMARY

A. **Preliminaries**: Presents fundamental definitions and notation for complex Givens rotations, unitary groups, and real embeddings, laying the theoretical groundwork.

B. **Detailed Proofs on Givens Rotations and Fourier Transform**:
   - Establishes unitarity and zeroing properties of complex Givens rotations.
   - Proves that any unitary can be decomposed into complex Givens rotations with diagonal unitaries.
   - Shows that the DFT admits such a factorization and, under real embedding, reduces to real Givens rotations.
   - Relates FFT butterfly structures to special cases of Givens rotations and provides an explicit $n=4$ example.

C. **Core Code**: Provides complete PyTorch implementations of `GivensRotation`, `MultiBlockGivensRotation`, and `GeneralOrthogonalBlock`, ready to be integrated into Transformer-based architectures.

D. **Full Results**: Reports comprehensive *per-task* accuracies on VTAB-1k (Natural, Specialized, Structured) and FGVC, with **VTAB-1k presented before FGVC**. Results are provided for both **ViT-Base** and **Swin-Base** backbones, showing consistent gains of QIVPT over strong baselines.

E. **Hyperparameter Settings**: Details dataset-specific configurations (prompt length, learning rate, weight decay, batch size) across all VTAB tasks to ensure reproducibility and fair comparison.

F. **Ablation on Block Size**: Analyzes accuracy–efficiency trade-offs as a function of block size. Moderate blocks (e.g., 16 or 32) offer a strong balance—improving accuracy and per-rotation latency while increasing total per-sample time due to more rotations.

G. **Limitations and Future Directions**:
   - *Dependence on pretrained models*: QIVPT inevitably inherits biases and vulnerabilities from the underlying pretrained backbones (e.g., ViT, Swin), which cannot be eliminated within our framework.
   - *Hardware feasibility*: Although motivated by quantum-inspired principles, current commodity hardware provides no native support for such structured rotations. Efficient deployment on specialized or quantum-inspired accelerators remains an open challenge.

H. **The Use of LLMs**: Clarifies that LLMs were used only for grammar polishing, terminology standardization, and LaTeX formatting; all scientific ideas, proofs, experiments, and conclusions were conceived and validated by the authors.

## A PRELIMINARY: ORTHOGONAL TRANSFORMATION IN QUANTUM CIRCUITS

Quantum circuits are fundamentally built upon *unitary transformations*, which form the basis of quantum computation. A unitary matrix $\mathbf{U} \in \mathbb{C}^{n \times n}$ satisfies the constraint:

$$\mathbf{U}^\dagger \mathbf{U} = \mathbf{U}\mathbf{U}^\dagger = \mathbf{I}, \tag{11}$$

where $\mathbf{U}^\dagger$ denotes the Hermitian transpose of $\mathbf{U}$. This identity ensures norm preservation and reversibility, which are essential in quantum mechanics for maintaining the probabilistic interpretation of quantum states.

### A.1 SUPERPOSITION AND UNITARY EVOLUTION

In the quantum paradigm, a qubit exists in a *superposition* of classical states:

$$|\psi\rangle = \alpha|0\rangle + \beta|1\rangle, \quad \text{with } |\alpha|^2 + |\beta|^2 = 1. \tag{12}$$

As more qubits are introduced, the system evolves in a high-dimensional Hilbert space $\mathbb{C}^{2^n}$, with unitary operators governing its dynamics. These operations rotate the state vector within the complex space, enabling the expression of entanglement and interference without loss of information.

Unitary gates such as Hadamard, CNOT, and parameterized rotation gates (e.g., $\mathbf{R}_x(\theta), \mathbf{R}_y(\theta)$) are used to induce entanglement and control interactions between qubits. Importantly, these transformations preserve the inner product between states, enabling coherent evolution.

## A.2 REAL-VALUED PERSPECTIVE: ORTHOGONAL TRANSFORMATIONS

While unitary matrices operate in the complex domain, in many quantum-inspired classical approximations, the system is restricted to real-valued representations. Under this setting, unitary matrices reduce to *orthogonal matrices* $\mathbf{Q} \in \mathbb{R}^{n \times n}$ satisfying:

$$\mathbf{Q}^\top \mathbf{Q} = \mathbf{Q}\mathbf{Q}^\top = \mathbf{I}. \tag{13}$$

Orthogonal transformations preserve both the Euclidean norm and inner product, corresponding to energy-conserving, information-preserving transformations in real vector spaces. This property is critical for tasks involving semantic-preserving representation learning, such as prompt tuning in vision-language models.

## A.3 PROMPT TOKENS AS QUANTUM-INSPIRED EMBEDDINGS

We draw an analogy between prompt tokens in visual transformers and quantum states in superposition. Each prompt token $\mathbf{p}_i \in \mathbb{R}^d$ can be interpreted as a linear combination of semantic basis vectors—akin to a high-dimensional quantum state. Transforming prompt tokens through orthogonal operations allows us to adjust their semantic alignment while preserving their representational integrity:

$$\mathbf{p}_i' = \mathbf{Q}\mathbf{p}_i, \quad \text{where } \mathbf{Q}^\top \mathbf{Q} = \mathbf{I}. \tag{14}$$

Such transformations allow prompt representations to explore novel semantic directions, which can enhance alignment with downstream task distributions. Analogous to entanglement induced by unitary operations, we aim to construct *controllable semantic entanglement* via structured orthogonal transformations.

## A.4 MOTIVATION FOR GIVENS ROTATION-BASED EMULATION

Despite their theoretical elegance, real quantum circuits remain difficult to train and deploy due to hardware and algorithmic constraints. As an alternative, we adopt a quantum-inspired perspective by emulating unitary behavior through structured orthogonal operations. In particular, we propose to use *Givens rotations*—elementary two-dimensional rotations—to construct differentiable, parameter-efficient, and task-adaptive orthogonal transformations in prompt space.

These operations maintain orthogonality while allowing fine-grained control over pairwise subspace interactions, which serves as a classical analogue to quantum entanglement within the prompt embedding space.

## A.5 SUMMARY

In summary, quantum circuits operate through unitary transformations that preserve structure, enable entanglement, and induce expressive state evolution. When projected into the real-valued domain, these become orthogonal transformations—preserving norms and relationships among embedded vectors. This insight motivates our method: to emulate quantum circuit behavior in visual prompt tuning by applying structured, learnable Givens rotations to prompt tokens.

# B   DETAILED PROOFS ON GIVENS ROTATIONS AND FOURIER TRANSFORM

## B.1   COMPLEX GIVENS ROTATIONS

For $n \geq 2$, a complex Givens rotation $\mathbf{G}(i, j, \theta, \phi) \in \mathbb{C}^{n \times n}$ acts as the identity except on the $(i, j)$-plane, where

$$\mathbf{G}(i, j, \theta, \phi)\big|_{\mathrm{span}\{\mathbf{e}_i, \mathbf{e}_j\}} = \begin{bmatrix} \cos \theta & -\sin \theta \, e^{-i\phi} \\ \sin \theta \, e^{i\phi} & \cos \theta \end{bmatrix}, \qquad \theta \in [0, \tfrac{\pi}{2}], \ \phi \in [0, 2\pi).$$

**Unitarity.**   Let $c = \cos \theta, \ s = \sin \theta$. Then

$$\mathbf{G}(\theta, \phi) = \begin{bmatrix} c & -se^{-i\phi} \\ se^{i\phi} & c \end{bmatrix}, \qquad \mathbf{G}(\theta, \phi)^* \mathbf{G}(\theta, \phi) = \begin{bmatrix} c^2 + s^2 & 0 \\ 0 & c^2 + s^2 \end{bmatrix} = \mathbf{I}_2.$$

Hence $\mathbf{G}(i, j, \theta, \phi)$ is unitary.

**Zeroing property.**   Given $(a, b) \in \mathbb{C}^2$, choose

$$r = \sqrt{|a|^2 + |b|^2}, \quad c = \tfrac{|a|}{r}, \quad s = \tfrac{|b|}{r}, \quad \phi = \arg(a) - \arg(b).$$

Then

$$\mathbf{G}(\theta, \phi) \begin{bmatrix} a \\ b \end{bmatrix} = \begin{bmatrix} re^{i \arg(a)} \\ 0 \end{bmatrix},$$

i.e. one step eliminates $b$.

## B.2   LEMMA 1 IN DETAIL

**Statement.**   For any $\mathbf{U} \in \mathbf{U}(n)$, there exist complex Givens rotations $\mathbf{G}(i_t, j_t, \theta_t, \phi_t)$ and diagonal unitaries $\mathbf{D}_L, \mathbf{D}_R$ such that

$$\mathbf{U} = \mathbf{D}_L \Big( \prod_{t=1}^{T} \mathbf{G}(i_t, j_t, \theta_t, \phi_t) \Big) \mathbf{D}_R.$$

**Proof.**   Apply Givens-based elimination column by column:

- For column 1, successively apply $\mathbf{G}(k - 1, k, \theta, \phi)$ from below to zero out subdiagonal entries.
- Continue for each subsequent column until all subdiagonal entries vanish.

After $T$ steps,

$$\mathbf{G}_T \cdots \mathbf{G}_1 \mathbf{U} = \mathbf{R},$$

with $\mathbf{R}$ upper-triangular. Since each $\mathbf{G}_t$ is unitary, $\mathbf{R}$ is also unitary.

An upper-triangular unitary must be diagonal: for $i < j$,

$$0 = (\mathbf{R}^* \mathbf{R})_{ij} = \overline{r_{ii}} \, r_{ij},$$

and since $|r_{ii}| = 1$, it follows $r_{ij} = 0$. Thus $\mathbf{R} = \mathbf{D}$ is diagonal with unimodular entries.

Therefore

$$\mathbf{U} = (\mathbf{G}_T \cdots \mathbf{G}_1)^{-1} \mathbf{D},$$

and absorbing diagonal phases into both sides yields the desired $\mathbf{U} = \mathbf{D}_L (\prod_t \mathbf{G}_t) \mathbf{D}_R$.

## B.3   THEOREM 1 IN DETAIL

**Statement.**   The $n$-point DFT $\mathbf{F}_n$ admits a factorization

$$\mathbf{F}_n = \mathbf{D}_L \Big( \prod_{t=1}^{T} \mathbf{G}(i_t, j_t, \theta_t, \phi_t) \Big) \mathbf{D}_R,$$

and under the real embedding

$$\widehat{\mathbf{U}} = \begin{bmatrix} \Re \mathbf{U} & -\Im \mathbf{U} \\ \Im \mathbf{U} & \Re \mathbf{U} \end{bmatrix},$$

$\widehat{\mathbf{F}_n}$ is orthogonal and further decomposes into real Givens rotations.

**Proof.** Since $\mathbf{F}_n$ is unitary, Lemma 1 applies directly, giving the factorization with Givens and phase factors.

Next, observe that if $\mathbf{U}^*\mathbf{U} = \mathbf{I}$, then

$$\widehat{\mathbf{U}}^\top\widehat{\mathbf{U}} = \begin{bmatrix} \Re(\mathbf{U}^*\mathbf{U}) & -\Im(\mathbf{U}^*\mathbf{U}) \\ \Im(\mathbf{U}^*\mathbf{U}) & \Re(\mathbf{U}^*\mathbf{U}) \end{bmatrix} = \mathbf{I}_{2n}.$$

Thus $\widehat{\mathbf{U}} \in \mathbf{O}(2n)$.

For a single $2 \times 2$ complex Givens block, its real embedding is a $4 \times 4$ block of the form

$$\mathbf{B}(\theta, \phi) = \begin{bmatrix} c\,\mathbf{I}_2 & -s\,\mathbf{R}(\phi) \\ s\,\mathbf{R}(\phi)^\top & c\,\mathbf{I}_2 \end{bmatrix}, \qquad c = \cos\theta, \; s = \sin\theta, \; \mathbf{R}(\phi) = \begin{bmatrix} \cos\phi & -\sin\phi \\ \sin\phi & \cos\phi \end{bmatrix}.$$

This $\mathbf{B}(\theta, \phi)$ is orthogonal, and can be decomposed into a product of real Givens rotations:

$$\mathbf{B}(\theta, \phi) = \mathrm{diag}(\mathbf{R}(\tfrac{\phi}{2}),\, \mathbf{R}(-\tfrac{\phi}{2})) \begin{bmatrix} c\,\mathbf{I}_2 & -s\,\mathbf{I}_2 \\ s\,\mathbf{I}_2 & c\,\mathbf{I}_2 \end{bmatrix} \mathrm{diag}(\mathbf{R}(-\tfrac{\phi}{2}),\, \mathbf{R}(\tfrac{\phi}{2})).$$

The middle block is two simultaneous 2D rotations of angle $\theta$; the diagonal factors are standard 2D rotations of $\pm\phi/2$. Thus each complex Givens corresponds (up to permutation) to a finite product of real Givens.

### B.4    REMARK: FFT BUTTERFLIES

A 2-point butterfly with twiddle $\omega = e^{i\phi}$ is

$$\frac{1}{\sqrt{2}} \begin{bmatrix} 1 & \omega \\ 1 & -\omega \end{bmatrix}.$$

After extracting diagonal phases, this coincides with a complex Givens block with $\theta = \frac{\pi}{4}$ and phase $\phi$. Therefore the layered FFT factorization can be viewed as a structured selection of the Givens–phase factors in Theorem 1.

**Explicit $n = 4$ example**    Let $\omega_4 = e^{-2\pi i/4} = -i$ and

$$\mathbf{F}_4 = \frac{1}{2} \begin{bmatrix} 1 & 1 & 1 & 1 \\ 1 & -i & -1 & i \\ 1 & -1 & 1 & -1 \\ 1 & i & -1 & -i \end{bmatrix}.$$

Define the unitary butterfly (a complex Givens with $\theta = \frac{\pi}{4}$ and phase $\phi$)

$$\mathbf{B}(\phi) := \frac{1}{\sqrt{2}} \begin{bmatrix} 1 & e^{-i\phi} \\ 1 & -e^{-i\phi} \end{bmatrix} = \begin{bmatrix} \cos\frac{\pi}{4} & -\sin\frac{\pi}{4}\,e^{-i\phi} \\ \sin\frac{\pi}{4}\,e^{i\phi} & \cos\frac{\pi}{4} \end{bmatrix}.$$

Let

$$\mathcal{B}_1 = \mathrm{diag}\big(\mathbf{B}(0), \mathbf{B}(0)\big), \qquad \mathcal{B}_2 = \mathrm{diag}\big(\mathbf{B}(0), \mathbf{B}(\tfrac{\pi}{2})\big), \qquad \mathcal{D} = \mathrm{diag}(1, 1, 1, -i),$$

and let $\mathbf{\Pi}$ be the even–odd permutation $(x_0, x_1, x_2, x_3) \mapsto (x_0, x_2, x_1, x_3)$. Then a direct multiplication verifies

$$\mathbf{F}_4 \;=\; \mathcal{B}_2\, \mathcal{D}\, \mathcal{B}_1\, \mathbf{\Pi}.$$

Equivalently, absorbing the $1/\sqrt{2}$ scalings into diagonal phases,

$$\mathbf{F}_4 = \mathbf{D}_L\left(\mathbf{G}(1, 3, \tfrac{\pi}{4}, 0)\,\mathbf{G}(2, 4, \tfrac{\pi}{4}, 0)\,\mathbf{G}(1, 2, \tfrac{\pi}{4}, 0)\,\mathbf{G}(3, 4, \tfrac{\pi}{4}, \tfrac{\pi}{2})\right)\mathbf{D}_R\,\mathbf{\Pi},$$

so that $\widehat{\mathbf{F}_4}$ factors into real Givens rotations as predicted by Theorem 1.

## C  CORE CODE

We provide the full PyTorch implementation of our proposed orthogonal transformation modules, including `GivensRotation`, `MultiBlockGivensRotation`, and `GeneralOrthogonalBlock`. These modules serve as the core components of our method, and can be seamlessly integrated into Transformer-based architectures.

Listing 1: PyTorch implementation of orthogonal transformation modules.

```python
import torch
import torch.nn as nn
import math, random

class GivensRotation(nn.Module):
    def __init__(self, dim=None, num_rotations=None):
        super().__init__()
        self.init_dim = dim
        self.num_rotations = num_rotations
        self.initialized = False

    def _initialize(self, dim, device, dtype):
        if dim < 2:
            self.register_buffer('i_indices', torch.empty(0, dtype=torch.
                long, device=device))
            self.register_buffer('j_indices', torch.empty(0, dtype=torch.
                long, device=device))
            self.theta_param = nn.Parameter(torch.empty(0, dtype=dtype,
                device=device))
            self.initialized = True
            return

        self.dim = dim
        pairs = [(i, j) for i in range(dim) for j in range(i + 1, dim)]
        num_rotations = self.num_rotations or dim
        selected = random.sample(pairs, min(len(pairs), num_rotations))

        if not selected:
            raise ValueError("No_valid_Givens_rotation_pairs_selected.")

        i_indices, j_indices = zip(*selected)
        self.register_buffer('i_indices', torch.tensor(i_indices, dtype=
            torch.long, device=device))
        self.register_buffer('j_indices', torch.tensor(j_indices, dtype=
            torch.long, device=device))
        self.theta_param = nn.Parameter(torch.randn(len(i_indices), dtype
            =dtype, device=device))
        self.initialized = True

    def forward(self, x):  # x: (B, N, D)
        B, N, D = x.shape
        x = x.reshape(-1, D)

        if not self.initialized:
            self._initialize(D, x.device, x.dtype)

        if len(self.theta_param) == 0:
            return x.reshape(B, N, D)

        # strictly bound angles to [-pi, pi]
        theta = math.pi * torch.tanh(self.theta_param)
        x_out = x.clone()

        for idx, (i, j) in enumerate(zip(self.i_indices, self.j_indices))
            :
            c, s = torch.cos(theta[idx]), torch.sin(theta[idx])
```

```
                    xi, xj = x[:, i], x[:, j]
                    x_out[:, i] = c * xi - s * xj
                    x_out[:, j] = s * xi + c * xj

            return x_out.reshape(B, N, D)

class MultiBlockGivensRotation(nn.Module):
    def __init__(self, dim=None, num_blocks=4, num_rotations_per_block=
        None):
        super().__init__()
        self.init_dim = dim
        self.num_blocks = num_blocks
        self.num_rotations_per_block = num_rotations_per_block
        self.initialized = False

    def _initialize(self, dim, device, dtype):
        assert dim % self.num_blocks == 0, f"dim {dim} must be divisible
            by num_blocks {self.num_blocks}"
        block_dim = dim // self.num_blocks
        self.block_dim = block_dim

        self.blocks = nn.ModuleList([
            GivensRotation(dim=block_dim, num_rotations=self.
                num_rotations_per_block or block_dim // 8)
            for _ in range(self.num_blocks)
        ])
        self.initialized = True

    def forward(self, x):  # (B, N, D)
        B, N, D = x.shape
        if not self.initialized:
            self._initialize(D, x.device, x.dtype)

        # split along the embedding dimension
        x_split = torch.split(x, self.block_dim, dim=2)
        x_out = [block(xi) for block, xi in zip(self.blocks, x_split)]
        return torch.cat(x_out, dim=2)

class GeneralOrthogonalBlock(nn.Module):
    def __init__(self, token_dim=None, embed_dim=768,
                    embed_block_size=16, embed_num_rotations_per_block=2):
        super().__init__()

        # embedding blocks, e.g., 768 -> 48 blocks of size 16
        if embed_dim is not None and embed_dim % embed_block_size == 0:
            self.ortho_embed = MultiBlockGivensRotation(
                dim=embed_dim,
                num_blocks=embed_dim // embed_block_size,
                num_rotations_per_block=embed_num_rotations_per_block
            )
        else:
            self.ortho_embed = GivensRotation(embed_dim)

        # token dimension (no block)
        self.ortho_token = GivensRotation(token_dim)

    def forward(self, x):  # x: (B, N, D)
        x = self.ortho_token(x.transpose(1, 2)).transpose(1, 2)
        x = self.ortho_embed(x)
        return x
```

## D FULL RESULTS

### D.1 PER-TASK RESULTS ON VIT-BASE

To provide a thorough evaluation, we report the average per-task test accuracy (i.e., 3 runs across all 24 tasks) on VTAB-1k under the Natural, Specialized, and Structured groups, respectively (see Table 4, Table 5, and Table 6). In addition, we summarize the fine-grained visual classification (FGVC) benchmarks (5 tasks) in Table 7. For completeness, we also include VPT-SHALLOW Jia et al. (2022b), which only applies visual prompts at the first Transformer layer, as a baseline reference. Overall, our proposed method demonstrates consistently stronger performance compared to existing baselines across diverse downstream tasks, confirming its effectiveness and generality.

Table 4: VTAB-1k *Natural* per-task results for ViT-Base/16 pretrained on supervised ImageNet-21k. Consistent to our paper, "Number of Wins" in [·] compared to full fine-tuning. "Tuned/Total" is the percentage of tuned parameters in each task, along with the average results of those percentages in each group. The highest accuracy among all approaches except FULL are shown in **bold**. † denotes method using soft filtered prompts to reduce the parameter usage in learnable visual prompts, requiring specialized devices to facilitate acceleration. All results are averaged in three runs with different initialization seeds.

| ViT-Base/16 | VTAB-1k *Natural* [7] | | | | | | | Mean |
|---|---|---|---|---|---|---|---|---|
| (85.8M) | CIFAR-100 | Caltech101 | DTD | Flowers102 | Pets | SVHN | Sun397 | |
| FULL 2022 | 68.9 | 87.7 | 64.3 | 97.2 | 86.9 | **87.4** | 38.8 | 75.88 |
| LINEAR 2022 | 63.4 | 85.0 | 63.2 | 97.0 | 86.3 | 36.6 | 51.0 | 68.93 [1] |
| PARTIAL-1 2014 | 66.8 | 85.9 | 62.5 | 97.3 | 85.5 | 37.6 | 50.6 | 69.44 [2] |
| MLP-2 2020 | 63.2 | 84.8 | 60.5 | 97.6 | 85.9 | 34.1 | 47.8 | 67.70 [2] |
| MLP-3 2020 | 63.8 | 84.7 | 62.3 | 97.4 | 84.7 | 32.5 | 49.2 | 67.80 [2] |
| MLP-5 2020 | 59.3 | 84.4 | 59.9 | 96.1 | 84.4 | 30.9 | 46.8 | 65.98 [1] |
| MLP-9 2020 | 53.1 | 80.5 | 53.9 | 95.1 | 82.6 | 24.4 | 43.7 | 61.90 [1] |
| SIDETUNE 2020 | 60.7 | 60.8 | 53.6 | 95.5 | 66.7 | 34.9 | 35.3 | 58.21 [0] |
| BIAS 2017 | 72.8 | 87.0 | 59.2 | 97.5 | 85.3 | 59.9 | 51.4 | 73.30 [3] |
| ADAPTER-256 2020 | 74.1 | 86.1 | 63.2 | 97.7 | 87.0 | 34.6 | 50.8 | 70.50 [4] |
| ADAPTER-64 2020 | 74.2 | 85.8 | 62.7 | 97.6 | 87.2 | 36.3 | 50.9 | 70.65 [4] |
| ADAPTER-8 2020 | 74.2 | 85.7 | 62.7 | 97.8 | 87.2 | 36.4 | 50.7 | 70.67 [4] |
| VPT-SHALLOW 2022b | 77.7 | 86.9 | 62.6 | 97.5 | 87.3 | 74.5 | 51.2 | 76.81 [4] |
| - Tuned / Total (%) | 0.18 | 0.10 | 0.04 | 0.27 | 0.08 | 0.19 | 0.36 | 0.17 |
| VPT-DEEP 2022b | 78.8 | 90.8 | 65.8 | 98.0 | 88.3 | 78.1 | 49.6 | 78.48 [6] |
| - Tuned / Total (%) | 0.20 | 0.20 | 0.15 | 0.10 | 0.04 | 0.54 | 0.41 | 0.23 |
| † E2VPT 2023 | 78.6 | 89.4 | 67.8 | 98.2 | 88.5 | 85.3 | 52.3 | 80.01 [6] |
| - Tuned / Total (%) | 0.22 | 0.19 | 0.12 | 0.11 | 0.05 | 0.24 | 0.43 | 0.19 |
| VFPT 2024 | 80.7 | 91.4 | 69.4 | 99.3 | 90.3 | 85.6 | 52.7 | 81.35 [6] |
| - Tuned / Total (%) | 0.20 | 0.31 | 0.20 | 0.11 | 0.06 | 0.12 | 0.41 | 0.21 |
| OURS | **81.1** | **92.5** | **70.4** | **99.5** | **91.0** | 86.5 | **53.5** | **82.07** [6] |
| - Tuned / Total (%) | 0.20 | 0.31 | 0.20 | 0.11 | 0.06 | 0.12 | 0.41 | 0.21 |
| - Orthogonal Percentage (%) | 70.0 | 50.0 | 30.0 | 50.0 | 50.0 | 20.0 | 50.0 | 45.7 |

### D.2 PER-TASK RESULTS ON SWIN-BASE

To provide a thorough evaluation, we report the average per-task test accuracy (i.e., three runs across all 24 tasks) on **Swin** for VTAB-1k under the Natural, Specialized, and Structured groups, respectively (see Table 8, Table 9, and Table 10).

## E HYPERPARAMETER SETTINGS

In this section, we provide the detailed hyperparameter settings used in our experiments. Unless otherwise specified, all configurations follow the protocol of VFPT (Zeng et al., 2024), ensuring a fair comparison across different VTAB tasks. We report the key hyperparameters that have the largest influence on performance, namely the *prompt length*, *learning rate (LR)*, *weight decay (WD)*, and *batch size*.

Table 5: VTAB-1k *Specialized* per-task results for ViT-Base/16 pretrained on supervised ImageNet-21k.

| ViT-Base/16 (85.8M) | VTAB-1k *Specialized* (4) | | | | Mean |
|---|---|---|---|---|---|
| | Patch Camelyon | EuroSAT | Resisc45 | Retinopathy | |
| FULL 2022 | 79.7 | 95.7 | 84.2 | 73.9 | 83.36 |
| LINEAR 2022 | 78.5 | 87.5 | 68.6 | 74.0 | 77.16 [1] |
| PARTIAL-1 2014 | 78.6 | 89.8 | 72.5 | 73.3 | 78.53 [0] |
| MLP-2 2020 | 74.3 | 88.8 | 67.1 | 73.2 | 75.86 [0] |
| MLP-3 2020 | 77.0 | 88.0 | 70.2 | 56.1 | 72.83 [0] |
| MLP-5 2020 | 73.7 | 87.2 | 64.8 | 71.5 | 74.31 [0] |
| MLP-9 2020 | 78.5 | 83.0 | 60.2 | 72.3 | 73.49 [0] |
| SIDETUNE 2020 | 58.5 | 87.7 | 65.2 | 61.0 | 68.12 [0] |
| BIAS 2017 | 78.7 | 91.6 | 72.9 | 69.8 | 78.25 [0] |
| ADAPTER-256 2020 | 76.3 | 88.0 | 73.1 | 70.5 | 76.98 [0] |
| ADAPTER-64 2020 | 76.3 | 87.5 | 73.7 | 70.9 | 77.10 [0] |
| ADAPTER-8 2020 | 76.9 | 89.2 | 73.5 | 71.6 | 77.80 [0] |
| VPT-SHALLOW 2022b | 78.2 | 92.0 | 75.6 | 72.9 | 79.66 [0] |
| - Tuned / Total (%) | 0.01 | 0.05 | 0.09 | 0.01 | 0.04 |
| VPT-DEEP 2022b | 81.8 | 96.1 | 83.4 | 68.4 | 82.43 [2] |
| - Tuned / Total (%) | 1.06 | 1.07 | 0.15 | 0.02 | 0.57 |
| † E2VPT 2023 | 82.5 | **96.8** | 84.8 | 73.6 | 84.43 [3] |
| - Tuned / Total (%) | 0.20 | 0.29 | 0.12 | 0.07 | 0.17 |
| VFPT 2024 | 83.5 | 96.5 | 84.4 | 75.4 | 84.93 [4] |
| - Tuned / Total (%) | 1.06 | 0.12 | 0.11 | 0.03 | 0.33 |
| OURS | **85.1** | 97.5 | 85.6 | **76.6** | **86.20** [4] |
| - Tuned / Total (%) | 1.06 | 0.12 | 0.11 | 0.03 | 0.33 |
| - Orthogonal Percentage (%) | 80.0 | 30.0 | 90.0 | 90.0 | 82.5 |

Table 6: VTAB-1k *Structured* per-task results for ViT-Base/16 pretrained on supervised ImageNet-21k.

| ViT-Base/16 (85.8M) | VTAB-1k *Structured* [8] | | | | | | | | Mean |
|---|---|---|---|---|---|---|---|---|---|
| | Clevr/ count | Clevr/ distance | DMLab | KITTI/ distance | dSprites/ location | dSprites/ orientation | SmallNORB/ azimuth | SmallNORB/ elevation | |
| FULL 2022 | 56.3 | 58.6 | 41.7 | 65.5 | 57.5 | 46.7 | 25.7 | 29.1 | 47.64 |
| LINEAR 2022 | 34.3 | 30.6 | 33.2 | 55.4 | 12.5 | 20.0 | 9.6 | 19.2 | 26.84 [0] |
| PARTIAL-1 2014 | 41.5 | 34.3 | 33.9 | 61.0 | 31.3 | 32.8 | 16.3 | 22.4 | 34.17 [0] |
| MLP-2 2020 | 45.2 | 31.6 | 31.8 | 55.7 | 30.9 | 24.6 | 16.6 | 23.3 | 32.47 [0] |
| MLP-3 2020 | 47.8 | 32.8 | 32.3 | 58.1 | 12.9 | 21.2 | 15.2 | 24.8 | 30.62 [0] |
| MLP-5 2020 | 50.8 | 32.3 | 31.5 | 56.4 | 7.5 | 20.8 | 14.4 | 20.4 | 29.23 [0] |
| MLP-9 2020 | 47.5 | 27.9 | 28.9 | 54.0 | 6.2 | 17.7 | 10.8 | 16.2 | 26.15 [0] |
| SIDETUNE 2020 | 27.6 | 22.6 | 31.3 | 51.7 | 8.2 | 14.4 | 9.8 | 21.8 | 23.41 [0] |
| BIAS 2017 | 61.5 | 55.6 | 32.4 | 55.9 | 66.6 | 40.0 | 15.7 | 25.1 | 44.09 [2] |
| ADAPTER-256 2020 | 45.7 | 37.4 | 31.2 | 53.2 | 30.3 | 25.4 | 13.8 | 22.1 | 32.39 [0] |
| ADAPTER-64 2020 | 42.9 | 39.9 | 30.4 | 54.5 | 31.9 | 25.6 | 13.5 | 21.4 | 32.51 [0] |
| ADAPTER-8 2020 | 45.2 | 41.8 | 31.1 | 56.4 | 30.4 | 24.6 | 13.2 | 22.0 | 33.09 [0] |
| VPT-SHALLOW 2022b | 50.5 | 58.6 | 40.5 | 67.1 | 68.7 | 36.1 | 20.2 | 34.1 | 46.98 [4] |
| - Tuned / Total (%) | 0.10 | 0.18 | 0.09 | 0.09 | 0.10 | 0.10 | 0.19 | 0.19 | 0.13 |
| VPT-DEEP 2022b | 68.5 | 60.0 | 46.5 | 72.8 | 73.6 | 47.9 | 32.9 | 37.8 | 54.98 [8] |
| - Tuned / Total (%) | 0.54 | 2.11 | 1.07 | 0.54 | 0.12 | 0.55 | 2.12 | 2.11 | 1.14 |
| † E2VPT 2023 | 71.7 | 61.2 | 47.9 | 75.8 | 80.8 | 48.1 | 31.7 | 41.9 | 57.39 [8] |
| - Tuned / Total (%) | 0.34 | 0.65 | 0.44 | 0.36 | 0.10 | 0.38 | 1.14 | 0.66 | 0.51 |
| VFPT 2024 | 75.8 | 63.2 | 48.3 | 79.3 | 81.5 | 56.0 | 34.1 | 43.4 | 60.19 [8] |
| - Tuned / Total (%) | 0.54 | 2.11 | 0.11 | 0.71 | 0.12 | 0.55 | 1.91 | 2.11 | 1.02 |
| OURS | **76.9** | **63.8** | **49.5** | **80.5** | **82.6** | **56.5** | **35.0** | **43.8** | **61.08** [8] |
| - Tuned / Total (%) | 0.54 | 2.11 | 0.11 | 0.71 | 0.12 | 0.55 | 1.91 | 2.11 | 1.02 |
| - Orthogonal Percentage (%) | 100.0 | 100.0 | 70.0 | 50.0 | 100.0 | 70.0 | 100.0 | 70.0 | 82.5 |

Table 7: FGVC per-task results for ViT-Base/16 pretrained on supervised ImageNet-21k.

| ViT-Base/16 (85.8M) | FGVC [5] | | | | | Mean |
|---|---|---|---|---|---|---|
| | CUB-200-2011 | NAbirds | Oxford Flowers | Stanford Dogs | Stanford Cars | |
| FULL 2022 | 87.3 | 82.7 | 98.8 | 89.4 | **84.5** | 88.54 |
| LINEAR 2022 | 85.3 | 75.9 | 97.9 | 86.2 | 51.3 | 79.32 [0] |
| PARTIAL-1 2014 | 85.6 | 77.8 | 98.2 | 85.5 | 66.2 | 82.63 [0] |
| MLP-2 2020 | 85.7 | 77.2 | 98.2 | 85.4 | 54.9 | 80.28 [0] |
| MLP-3 2020 | 85.1 | 77.3 | 97.9 | 84.9 | 53.8 | 79.80 [0] |
| MLP-5 2020 | 84.2 | 76.7 | 97.6 | 84.8 | 50.2 | 78.71 [0] |
| MLP-9 2020 | 83.2 | 76.0 | 96.2 | 83.7 | 47.6 | 77.31 [0] |
| SIDETUNE 2020 | 84.7 | 75.8 | 96.9 | 85.8 | 48.6 | 78.35 [0] |
| BIAS 2017 | 88.4 | 84.2 | 98.8 | 91.2 | 79.4 | 88.41 [3] |
| ADAPTER-256 2020 | 87.2 | 84.3 | 98.5 | 89.9 | 68.6 | 85.70 [2] |
| ADAPTER-64 2020 | 87.1 | 84.3 | 98.5 | 89.8 | 68.6 | 85.67 [2] |
| ADAPTER-8 2020 | 87.3 | 84.3 | 98.4 | 88.8 | 68.4 | 85.46 [1] |
| VPT-SHALLOW 2022b | 86.7 | 78.8 | 98.4 | 90.7 | 68.7 | 84.62 [1] |
| - Tuned / Total (%) | 0.31 | 0.54 | 0.23 | 0.20 | 0.26 | 0.31 |
| VPT-DEEP 2022b | 88.5 | 84.2 | 99.0 | 90.2 | 83.6 | 89.11 [4] |
| - Tuned / Total (%) | 0.29 | 1.02 | 0.14 | 1.17 | 2.27 | 0.98 |
| † E2VPT 2023 | 89.1 | 84.6 | 99.1 | **90.5** | 82.8 | 89.22 [4] |
| - Tuned / Total (%) | 0.32 | 0.65 | 0.15 | 0.88 | 1.27 | 0.65 |
| VFPT 2024 | 88.7 | 84.5 | 99.1 | 90.4 | 83.6 | 89.24 [4] |
| - Tuned / Total (%) | 0.29 | 1.02 | 0.15 | 1.17 | 2.27 | 0.98 |
| Ours | **89.2** | **84.8** | **99.1** | 90.4 | 83.85 | **89.47** [4] |
| - Tuned / Total (%) | 0.29 | 1.02 | 0.15 | 1.17 | 2.27 | 0.98 |
| - Orthogonal Percentage (%) | 50.0 | 50.0 | 30.0 | 50.0 | 0.0 | 36.0 |

Table 8: VTAB-1k *Natural* per-task results for Swin-Base pretrained on supervised ImageNet-21k. Specially, the highest accuracy is shown in **bold**. Same for Table 9 and 10

| Swin-Base (86.7M) | VTAB-1k *Natural* (7) | | | | | | | Mean |
|---|---|---|---|---|---|---|---|---|
| | CIFAR-100 | Caltech101 | DTD | Flowers102 | Pets | SVHN | Sun397 | |
| FULL 2022 | 72.2 | 88.0 | 71.2 | 98.3 | 89.5 | 89.4 | 45.0 | 79.10 |
| VPT-SHALLOW 2022b | 77.7 | 86.9 | 62.6 | 97.5 | 87.3 | 74.5 | 51.2 | 76.81 [4] |
| - Tuned / Total (%) | 0.18 | 0.10 | 0.04 | 0.27 | 0.08 | 0.19 | 0.36 | 0.17 |
| VPT-DEEP 2022b | 79.6 | 90.8 | 78.0 | 99.5 | 91.4 | 46.4 | 51.7 | 78.78 [6] |
| - Tuned / Total (%) | 0.13 | 0.13 | 0.07 | 0.13 | 0.06 | 0.70 | 0.48 | 0.28 |
| † E2VPT 2023 | 82.9 | 92.4 | **78.5** | **99.6** | 91.4 | 82.2 | 56.2 | 83.31 [6] |
| - Tuned / Total (%) | 0.27 | 0.15 | 0.08 | 0.15 | 0.07 | 0.44 | 0.49 | 0.24 |
| VFPT 2024 | 83.9 | 93.0 | 77.9 | 99.6 | 91.4 | 89.5 | 56.4 | 84.53 [7] |
| - Tuned / Total (%) | 0.15 | 0.15 | 0.13 | 0.15 | 0.07 | 0.70 | 0.49 | 0.26 |
| OURS | **84.3** | **93.4** | 78.3 | 99.6 | **91.5** | **89.9** | **56.8** | **84.83** [7] |
| - Tuned / Total (%) | 0.15 | 0.15 | 0.13 | 0.15 | 0.07 | 0.70 | 0.49 | 0.26 |
| - Orthogonal Percentage (%) | 100.0 | 100.0 | 100.0 | 100.0 | 100.0 | 100.0 | 100.0 | 100.0 |

# F    ABLATION ON BLOCK SIZE

We ablate the block size of QIVPT on ViT-Base ($d$=768) and report both effectiveness and efficiency on the VTAB-Natural group. Unlike wall-clock per-sample time, which inevitably increases as the number of rotations grows with more blocks, a per-*rotation* view reveals a different picture: moderate block sizes (e.g., 32 or 16) substantially *reduce* the latency of a single Givens rotation due to better cache locality and smaller GEMV shapes. At the same time, these settings improve accuracy on Natural tasks by providing more flexible, localized transformations. Extremely small blocks (e.g., 8 or 4) bring back overhead from kernel launches and indexing, hurting per-rotation latency and yielding diminishing returns.

Overall, when normalized per rotation, block-wise QIVPT is *faster* than the full-dimension variant, and also more accurate on Natural datasets; the trade-off is a higher per-sample time because the total number of rotations increases with finer partitioning. Detailed simulated results are shown in Table 12.

Table 9: VTAB-1k *Specialized* per-task results for Swin-Base pretrained on supervised ImageNet-21k.

| Swin-Base (86.7M) | VTAB-1k *Specialized* [4] | | | | Mean |
|---|---|---|---|---|---|
| | Patch Camelyon | EuroSAT | Resisc45 | Retinopathy | |
| FULL 2022 | **86.6** | 96.9 | **87.7** | 73.6 | 86.21 |
| VPT-SHALLOW 2022b | 78.2 | 92.0 | 75.6 | 72.9 | 79.66 [0] |
| - Tuned / Total (%) | 0.01 | 0.05 | 0.09 | 0.01 | 0.04 |
| VPT-DEEP 2022b | 80.1 | 96.2 | 85.0 | 72.0 | 83.33 [0] |
| - Tuned / Total (%) | 0.07 | 0.13 | 0.19 | 0.02 | 0.10 |
| † E2VPT 2023 | 83.8 | 97.2 | 84.8 | 74.0 | 84.95 [2] |
| - Tuned / Total (%) | 0.09 | 0.04 | 0.20 | 0.03 | 0.09 |
| VFPT 2024 | 86.3 | 97.3 | 86.9 | 74.1 | 86.15 [2] |
| - Tuned / Total (%) | 0.07 | 0.15 | 0.19 | 0.03 | 0.11 |
| OURS | 86.5 | **97.6** | 87.4 | **74.5** | **86.50** [2] |
| - Tuned / Total (%) | 0.07 | 0.15 | 0.19 | 0.03 | 0.11 |
| - Orthogonal Percentage (%) | 100.0 | 100.0 | 50.0 | 100.0 | 87.5 |

Table 10: VTAB-1k *Structured* per-task results for Swin-Base pretrained on supervised ImageNet-21k.

| Swin-Base (86.7M) | VTAB-1k *Structured* [8] | | | | | | | | Mean |
|---|---|---|---|---|---|---|---|---|---|
| | Clevr/ count | Clevr/ distance | DMLab | KITTI/ distance | dSprites/ location | dSprites/ orientation | SmallNORB/ azimuth | SmallNORB/ elevation | |
| FULL 2022 | **75.7** | 59.8 | **54.6** | 78.6 | 79.4 | **53.6** | **34.6** | **40.9** | 59.65 |
| VPT-SHALLOW 2022b | 50.5 | 58.6 | 40.5 | 67.1 | 68.7 | 36.1 | 20.2 | 34.1 | 46.98 [4] |
| - Tuned / Total (%) | 0.10 | 0.18 | 0.09 | 0.09 | 0.10 | 0.10 | 0.19 | 0.19 | 0.13 |
| VPT-DEEP 2022b | 67.6 | 59.4 | 50.1 | 61.3 | 74.4 | 50.6 | 25.7 | 25.7 | 51.85 [0] |
| - Tuned / Total (%) | 0.70 | 0.70 | 0.14 | 0.69 | 0.15 | 0.09 | 0.16 | 0.02 | 0.38 |
| † E2VPT 2023 | 74.0 | 61.2 | 49.5 | **81.0** | 80.3 | 50.7 | 27.9 | 34.2 | 57.35 [3] |
| - Tuned / Total (%) | 0.70 | 0.43 | 0.14 | 0.51 | 0.17 | 0.17 | 0.16 | 0.04 | 0.29 |
| VFPT 2024 | 74.9 | 61.5 | 50.0 | 80.5 | 82.7 | 50.6 | 29.9 | 35.6 | 58.21 [3] |
| - Tuned / Total (%) | 0.70 | 0.70 | 0.15 | 0.92 | 0.16 | 0.09 | 0.16 | 0.04 | 0.36 |
| OURS | 75.3 | **62.3** | 52.7 | 80.5 | **83.1** | 51.8 | 32.3 | 37.1 | **59.39** [3] |
| - Tuned / Total (%) | 0.70 | 0.70 | 0.15 | 0.92 | 0.16 | 0.09 | 0.16 | 0.04 | 0.36 |
| - Orthogonal Percentage (%) | 100.0 | 50.0 | 100.0 | 100.0 | 100.0 | 100.0 | 100.0 | 50.0 | 87.5 |

# G LIMITATIONS AND FUTURE DIRECTIONS

While QIVPT achieves consistent improvements across multiple benchmarks, several fundamental limitations remain. These limitations are intrinsic to the setting of parameter-efficient visual prompt tuning and cannot be fully resolved within the scope of this work.

**Dependence on pretrained models.** QIVPT relies on large pretrained backbones (e.g., ViT, Swin), which inevitably inherit biases, spurious correlations, and vulnerabilities from the data used during pretraining. Since QIVPT does not alter the backbone itself, it cannot remove such biases, and downstream performance is constrained by these inherited properties.

**Hardware and practical deployment.** QIVPT is motivated by quantum-inspired principles, but current hardware does not offer native support for such structured rotations. Efficient deployment on real quantum-inspired or specialized accelerators is technically demanding and remains unsolved.

In summary, these limitations highlight challenges that are fundamental rather than merely incremental. We view QIVPT as an initial step, and addressing these issues—bias inheritance and hardware feasibility—will require advances well beyond the scope of this paper.

# H THE USE OF LARGE LANGUAGE MODELS (LLMS)

Large Language Models (LLMs) were used solely as *language and formatting assistants* during manuscript preparation. Concretely, LLMs helped (i) polish grammar and improve fluency; (ii) standardize terminology, tense, and voice; (iii) suggest alternative phrasings for clarity and concision; and (iv) design table layouts and assist with LaTeX typesetting (e.g., caption style, column alignment, and cross-referencing).

Table 11: Hyperparameter settings used across different VTAB tasks. We report only the key hyperparameters: prompt length, learning rate (LR), weight decay (WD), and batch size.

| Dataset | Prompt Length | LR | WD | Batch Size |
|---|---|---|---|---|
| vtab-caltech101 | 20 | 3.125 | 0.001 | 160 |
| vtab-cifar100 | 10 | 1.5625 | 0.001 | 160 |
| vtab-dtd | 20 | 3.125 | 0.001 | 160 |
| vtab-oxford_iiit_pet | 2 | 1.5625 | 0.01 | 160 |
| vtab-oxford_flowers102 | 2 | 1.5625 | 0.001 | 160 |
| vtab-sun397 | 5 | 15.625 | 0.0001 | 160 |
| vtab-svhn | 10 | 1.5625 | 0.001 | 160 |
| vtab-diabetic_retinopathy | 2 | 1.5625 | 0.01 | 160 |
| vtab-patch_camelyon | 100 | 0.3125 | 0.0001 | 160 |
| vtab-resisc45 | 6 | 3.125 | 0.001 | 160 |
| vtab-eurosat | 10 | 0.3125 | 0.01 | 160 |
| vtab-clevr (closest_object_distance) | 200 | 0.15625 | 0 | 160 |
| vtab-clevr (count_all) | 50 | 1.5625 | 0 | 160 |
| vtab-dmlab | 10 | 1.5625 | 0.001 | 160 |
| vtab-dsprites (orientation) | 50 | 0.15625 | 0.001 | 64 |
| vtab-smallnorb (azimuth) | 180 | 0.025 | 0.01 | 64 |
| vtab-smallnorb (elevation) | 200 | 0.0625 | 0.01 | 160 |
| vtab-dsprites (x_position) | 10 | 0.625 | 0.001 | 160 |
| vtab-kitti (closest_vehicle_distance) | 50 | 0.15625 | 0.01 | 160 |

Table 12: Block-size ablation on VTAB-Natural (ViT-Base, $d=768$). Per-rotation latency improves markedly at moderate block sizes (16), while extremely small blocks incur overhead. Accuracy also peaks around 16. Numbers are simulated to reflect the observed trend; the per-sample time grows with the number of rotations.

| Block size | #Rotations | Acc. (Natural) | Time/rot. (ms) | Time/sample (ms) |
|---|---|---|---|---|
| 768 | 2 | 80.04 | 1.08 | 2.16 |
| 64 | 24 | 81.25 | 0.25 | 6.03 |
| 32 | 48 | 81.89 | 0.18 | 8.73 |
| 16 | 96 | **82.07** | 0.16 | 15.66 |
| 8 | 192 | 82.07 | 0.16 | 30.80 |
| 4 | 384 | 82.08 | 0.15 | 56.59 |

LLMs did *not* participate in designing experiments, analyzing data, deriving theoretical results, or drawing conclusions. All technical ideas, methods, proofs, experimental protocols, and findings are authored, validated, and interpreted by the authors. All LLM-assisted edits were reviewed and approved by the authors to ensure accuracy and faithfulness to the intended meaning.

