# OpenReview forum: "QIVPT: Quantum-Inspired Visual Prompt Tuning via Givens Rotations-Based Orthogonal Transformation"
_ICLR.cc/2026/Conference — ICLR 2026 Conference Withdrawn Submission_

### Official Review · Reviewer_3fP2 · 2025-10-22

**Soundness:** 3
**Presentation:** 2
**Contribution:** 2
**Rating:** 2
**Confidence:** 4

**Summary:**

This paper proposes rotating prompt tokens in Visual Prompt Tuning (VPT) through a designed orthogonal transformation. Experiments demonstrate that this approach achieves significant performance gains over the vanilla VPT method on VTAB-1K and FGVC benchmarks, utilizing ImageNet-21K pre-trained ViT and Swin backbones.

**Strengths:**

1. The introduction of an orthogonal transformation to rotate prompt tokens is a novel contribution. It intelligently increases inter-prompt discrimination while preserving the stability of their gradients.

2. The proposed method delivers strong empirical results, consistently outperforming the vanilla VPT baseline on standard VTAB-1K and FGVC benchmarks.

**Weaknesses:**

1.**Unclear motivations**. This work motivated from three points (cf. Line 192-197), but they require further elaboration:

 - Firstly, the authors claim that `"VPT’s interaction mechanism lacks structural guidance."` It is essential to clarify what is meant by "structural guidance" and explain its specific role in prompt tuning.

 - Secondly, the authors state that `"the reliance on a fixed embedding basis introduces a coordinate bias, which may restrict expressiveness and hinder generalization to unseen domains."` The term "fixed embedding basis" needs a precise definition. The authors should further explain how it introduces a coordinate bias and why this bias would restrict model expressiveness and hinder generalization. Additionally, does the proposed method itself demonstrate improved generalization to unseen domains?

 - Thirdly, the authors argue that `"the influence of prompts on attention dynamics and downstream predictions remains opaque, making the adaptation process difficult to interpret or control."` However, prior work [R1] has already explicitly detailed how prompts interact with images in the attention mechanism (e.g., in its Equation 3). The claim of opaqueness seems to overlook this existing explanation and needs to be justified.

  [R1] Attention to the Burstiness in Visual Prompt Tuning. ICCV 2025.

2. **Lack of explanation for performance gain**. The authors need to provide an explanation for why rotating the prompts leads to a significant improvement in accuracy. A comparative analysis of the relationships between prompts before and after rotation would be highly valuable to illustrate the mechanism behind the performance gain.

3. **Insufficient experimental validation**. The experimental setup is somewhat limited for a comprehensive assessment. Firstly, the work relies solely on supervised pre-trained backbones; inclusion of models trained with self-supervised paradigms (e.g., MAE, MoCo, DINO) is necessary to demonstrate broader applicability. Secondly, experiments are confined to image classification. Could the proposed method be effectively extended to other vision tasks, such as object detection or segmentation? Prompt length is a key design in prompt tuning methods. Could the authors provide an ablation study on the impact of prompt length?

4. **Missing shallow variant**. The proposed method applies orthogonal transformations at every transformer layer, constituting a "deep" variant analogous to VPT-Deep. How does the designed method perform in a shallow configuration?

**Questions:**

see weaknesses.

---

### Official Review · Reviewer_TWTn · 2025-10-24

**Soundness:** 3
**Presentation:** 2
**Contribution:** 2
**Rating:** 4
**Confidence:** 4

**Summary:**

The paper proposes a novel method called Quantum-Inspired Visual Prompt Tuning (QIVPT), which designs a structured and orthogonality-preserving transformation over prompts using learnable sequences of Givens rotations. The results show that QIVPT can consistently outperform existing prompt tuning baselines.

**Strengths:**

1. The paper is easy to follow, and the intro of Givens rotations does not involve additional training parameters.

2. Some results are promising on VTAB-1k and FGVC benchmarks.

**Weaknesses:**

1. The necessity and novelty of introducing Givens rotations are incremental. Specifically, Givens rotations, as claimed by the authors, can include the Fourier transformation. However, in relatively complex tasks (e.g., FGVC) and different architecture designs (e.g., Swin), the performance increase seems marginal.

2. The additional discussions on Sec 3.3 are a little bit redundant. In Sec. 3.2, the authors have already included the discussions on Givens rotations (Eqs. 5-6). The author can merge Sec. 3.3 into 3.2 for simplicity.

3. The discussion on interpretability is questionable. I understand that the authors followed the spirit of VFPT for interpretability discussions. However, in VFPT, more discussions are included in optimization landscapes, etc. In this paper, however, the only visualization results are from GradCAM. Besides, can the authors explain more, better in theory, that this can be defined as an improved interpretability? From my perspective, without including linearity, case-based reasoning, causal reasoning, or other similar approaches [1-4], it is hard to claim an improvement in this field.

4. The baselines can be improved. The latest paper in the comparison is from 2024. New approaches, such as [5-7] can be included for completeness.

[1] When do prompting and prefix-tuning work? a theory of capabilities and limitations

[2] Visual recognition with deep nearest centroids

[3] Causal abstractions of neural networks

[4] Re-Imagining Multimodal Instruction Tuning: A Representation View

[5] Visual instance-aware prompt tuning

[6] Diff-Prompt: Diffusion-Driven Prompt Generator with Mask Supervision

[7] CVPT: Cross Visual Prompt Tuning

**Questions:**

My major concern lies in the discussion of interpretability. It seems that the current discussion is insufficient and overclaimed. The authors need to explain more about why the proposed method has advanced explainability.

---

### Official Review · Reviewer_jkHG · 2025-11-01

**Soundness:** 2
**Presentation:** 3
**Contribution:** 3
**Rating:** 4
**Confidence:** 3

**Summary:**

This paper proposes QIVPT, a framework for adapting pre-trained Vision Transformers through structured orthogonal transformations inspired by quantum mechanics. Unlike conventional VPT, which treats prompt tokens as unconstrained parameters, QIVPT introduces a Givens Rotation Layer that models prompt evolution via sequences of learnable orthogonal transformations. Experiments on VTAB-1k and FGVC benchmarks show that QIVPT outperforms VPT and Fourier-based prompt tuning (VFPT).

**Strengths:**

- The paper introduces a unique quantum-inspired framework for prompt tuning, which is novel.

- Extensive experiments show consistent improvements across diverse benchmarks and architectures (ViT-B/16 and Swin-B), validating both accuracy and parameter-efficiency

- Includes comprehensive ablations on prompt depth, orthogonal ratio, and dimensionality, clarifying the design choices and their effects.

**Weaknesses:**

- Lack of clear interpretability benefit:
Although the paper claims improved interpretability by preserving orthogonality and geometric structure, it remains unclear what specific insight this provides in practice. Since individual prompts do not have explicit semantic meaning, understanding their pairwise “rotations” offers little tangible interpretability or diagnostic value for model behavior.

- Unclear motivation for semantic entanglement:
The notion of “semantic entanglement” among prompts is conceptually interesting but insufficiently justified. The paper does not clearly explain why such entanglement should improve task adaptation or generalization in visual prompt tuning.

- Weak rationale for using rotations as the mechanism:
It is not evident why orthogonal rotations are a necessary or uniquely effective means to induce prompt interactions. Other structured or parameter-efficient transformations (e.g., linear coupling, attention-based mixing) could in principle achieve similar effects. The paper’s argument for adopting Givens rotations as the preferred design is thus largely aesthetic and lacks empirical necessity.

- Practical Impact Ambiguity: The “quantum-inspired” aspect is conceptually appealing but remains metaphorical rather than operational—there is no real connection to quantum computation efficiency or simulation benefits.

- Limited Comparison with Recent VPT Variants: The paper compares against VFPT and E2VPT but omits newer or more diverse prompt-based methods such as [ref1, ref2, ref3, ref4]

[ref1] DA‑VPT: Semantic‑Guided Visual Prompt Tuning for Vision Transformers, CVPR 2025
[ref2] Visual Instance-aware Prompt Tuning, MM 2025
[ref3] Correlative and Discriminative Label Grouping for Multi-Label Visual Prompt Tuning, CVPR 2025
[ref4] Improving visual prompt tuning for selfsupervised vision transformers, ICML 2023

- Fairness of Parameter Control: The number of tuned parameters (tuned/total) varies notably among methods; e.g., E2VPT use different prompt counts, which may make comparisons less fair. The authors should equalize prompt counts or analyze scaling effects.

**Questions:**

- Can the proposed method be applied to self-supervised pre-trained ViT models?

---

### Official Review · Reviewer_e4xL · 2025-11-03

**Soundness:** 3
**Presentation:** 3
**Contribution:** 3
**Rating:** 6
**Confidence:** 4

**Summary:**

The paper proposes QIVPT, a parameter-efficient visual prompt tuning (VPT) technique that applies learnable sequences of givens rotations to prompt embeddings. Rotations are orthogonal (norm-preserving), angles are bounded via tanh, and transformations are applied per transformer layer, often in block-wise fashion for scalability.

**Strengths:**

- The overall contribution is interesting. Authors present clear, modular method. It drops into VPT pipelines and the contributions are well motivated.

- Strong empirical results on multiple benchmarks. The results on VTAB-1k and FGVC are strong and consistent.

- Authors provide core code and hyperparameters in appendix which make this work easy to be reproduced.

- The writing is cleared and smooth.

**Weaknesses:**

- Methodology:  The math shows existence of a Givens decomposition of the DFT, but the paper does not demonstrate a concrete initialization that recovers VFPT within the block-wise architecture actually used.

- Visualization: The orthogonality preserves geometry, but the paper doesn’t analyze learned rotations to show semantic structure (only Grad-CAM seems not enough, may plot the angle distributions or show which index pairs are used often.)

- Effciency: The authors uses simulated numbers, not measured runs. Should replace simulated times with real measurements. Report images/sec, wall-clock time per epoch, inference latency, and peak memory.

- Authors should review/ablation to the number of rotations K per block/layer. Please explain how K scales with dimension or block size. Are i, j pairs are fixed, learned or resampled?

**Questions:**

How many rotations per block and per layer are used in the main results? Are the index pairs fixed, learned, or resampled?

Can you share real training and inference throughput and memory for QIVPT vs VFPT and VPT on a key VTAB task?

Can you show a constructive setup that exactly recovers VFPT within QIVPT, with and without blocks?

---

### Author Response · Authors · 2025-11-22

We thank all reviewers for their constructive feedback and for recognizing the research value and innovative contributions of our work. We will further refine the paper by incorporating their suggestions, including improving the clarity of exposition and adding more comprehensive experimental results.

---

### Note · Authors · 2025-11-22

**Comment:**

We thank all reviewers for their constructive feedback and for recognizing the research value and innovative contributions of our work. We will further refine the paper by incorporating their suggestions, including improving the clarity of exposition and adding more comprehensive experimental results.

**Withdrawal Confirmation:**

I have read and agree with the venue's withdrawal policy on behalf of myself and my co-authors.